# Chitosan Extraction from *Goliathus orientalis* Moser, 1909: Characterization and Comparison with Commercially Available Chitosan

**DOI:** 10.3390/biomimetics5020015

**Published:** 2020-04-26

**Authors:** Pauline Fournier, Caroline R. Szczepanski, René-Paul Godeau, Guilhem Godeau

**Affiliations:** 1Institut de Physique de Nice (INPHYNI), Université Côte d’Azur, UMR 7010, 06000 Nice, France; pauline.fourniernice@gmail.com (P.F.); rp.godeau@club-internet.fr (R.-P.G.); 2Institut Méditerranéen du Risque de l’Environnement et du Développement Durable (IMREDD), Université Côte d’Azur, 06200 Nice, France; 3Department of Chemical Engineering & Materials Science, Michigan State University, East Lansing, MI 48824, USA; szcz@msu.edu

**Keywords:** goliath beetle, biopolymer extraction, chitin, chitosan

## Abstract

Chitosan is a polymer obtained by deacetylation of chitin, and chitin is one of the major components of the arthropod cuticle. Chitin and chitosan are both polysaccharides and are considered to be an interesting class of biosourced materials. This is evident as chitosan has already demonstrated utility in various applications in both industrial and biomedical domains. In the present work, we study the possibility to extract chitin and prepare chitosan from the Goliath beetle *Goliathus orientalis* Moser. The presented work includes description of this process and observation of the macroscopic and microscopic variations that occur in the specimen during the treatment. The prepared chitosan is characterized and compared with commercially available chitosan using infrared and thermogravimetric analysis. The deacetylation degree of prepared chitosan is also evaluated and compared with commercially available shrimp chitosan.

## 1. Introduction

Chitin, along with cellulose, is one of the most abundant biopolymers present in nature. Chitin is a major constituent of the Arthropoda’s shield as well as the cell wall of fungi. Furthermore, chitin is produced on a massive scale from the waste associated with processing of seafood products such as crab and shrimp [1,2,3,4,5]. Chemically speaking, chitin can be described as a linear polymer (polysaccharide) of β-1,4 linked N-acetylglucosamine, and different sub-types of chitin are described in the literature, including α, β and γ. These sub-types are determined by the organization of the polysaccharide fibres (e.g., parallel or anti-parallel). In Arthropoda’s exoskeleton, the most abundant form of chitin is the α form [6], which has increased stability due to a fully anti-parallel organization that leads to strong hydrogen interactions [7]. As a chemical, α-chitin is difficult to use, due to its poor solubility in most common, organic solvents, which results from strong intermolecular interactions and a compact macromolecular structure. Typically, to avoid this solubility limitation, chitin is modified prior to industrial use. The modified version of chitin that is more frequently employed is partially deacetylated; if this modification results in a high deacetylation degree (e.g., greater than 50% deacetylation) the material is typically referred to as chitosan. [8] Due to the free amine groups released during deacetylation, chitosan has a completely different solubility compared to chitin. As one example, chitosan is fully soluble in acidic water (whereas chitin is insoluble). Due to these favourable properties—biocompatibility, improved solubility, and biodegradability—chitosan has a wide range of applications in food, medical, and textile applications, as reported in reviews [9,10,11,12,13,14,15,16].

Among the various uses of chitosan, biologically related applications are of special relevance and currently a growing area of research. Chitosan is used in tissue engineering, the development of injectable hydrogels and wound healing as well as for implant materials [17,18,19,20,21,22]. In acidic conditions, chitosan has a positively charged structure. This characteristic causes chitosan to strongly interact with negatively charged entities including biomolecules such as lipids, proteins and nucleic acid derivatives. These interactions make chitosan very attractive for biomimetic drug delivery and biological applications [23,24,25]. Another growing area of applications for chitosan is in the food packaging industry. As an example, chitosan is often explored for the development of antibacterial packaging films that can avoid bacterial growth and/or kill microorganisms [26,27,28]. Beyond these more biologically focused applications, chitosan is also explored for metal complexion for catalysis or depollution, demonstrating the broad interests and uses of this molecule [29,30,31].

Chitosan’s solubility and other properties are strongly correlated with deacetylation degree, and, depending on the species used for chitin extraction, this degree can vary dramatically. As an example, Marei et al. reported deacetylation degrees of chitosan from various species: 74% for shrimp, 98% for locust, 96% for honeybee and 95% for beetle [32]. For industrial production and use, the majority of chitin is sourced from seafood waste (e.g., shrimp, crab, and lobster) but alternative chitin sources such as coral and fungus have also been reported [33,34,35,36] In the literature, various insects have also been investigated for chitin extraction and chitosan preparation, including beetles of various sizes, larvae, and adults [37,38,39,40,41], and, therefore, beetle cuticles may be considered a valuable source of chitin and chitosan for future industrial production. Beetles have long been considered a potential source of protein to feed humanity, since insect proteins are of good quality and their production requires minimal space and water, and also produces fewer greenhouse gases compared to production of classical animal proteins [42,43]. However, similar to seafood, large scale beetle consumption will produce waste [44], and, just as for shrimp, lobster and crab, the waste from beetle protein production will require treatment. One possibility to create value from such industrial waste is to extract chitin and chitosan. In this work, we investigate the possibility to use large-bodied beetles as specimens of interest for chitin extraction and chitosan preparation. 

Recently, our work has focused on the application of beetles as inspiration for materials science, with an emphasis on surface properties of beetle shells [45,46]. As one example, in a prior study we investigated the surface properties of *G. orientalis* Moser [47] and observed variations in surface wettability depending on the local coloration (white vs. black). In the present work, we shift our focus to the possibility of extracting chitin and preparing chitosan from *G. orientalis* Moser. Chitosan is obtained following a demineralization/deproteination/deacetylation strategy (Figure 1), and the products from this process are investigated and compared to commercially available shrimp chitosan using infrared and thermogravimetric analysis. For all modification steps, the *G. orientalis* exoskeletons were observed for macroscopic and microscopic morphologies. These observations highlight an interesting macroscopic evolution during the treatment, revealing hidden *G. orientalis* iridescence. This work confirms the possible use of *G. orientalis* as potential starting material for chitin extraction and chitosan preparation.

## 2. Materials and Methods

Specimens were acquired in a dead and dry state from a laboratory stock collection. All chemicals and solvent employed in this study were purchased from Sigma-Aldrich and used without further purification.

### 2.1. Chitin Extraction

Step 1, demineralization: dry *G. orientalis* (10.5 g) was hydrated in an aqueous 1M HCl solution. The solution was then warmed for 2 h (95 °C). The liquid phase was then removed and *G. orientalis* was rinsed with water until reaching a neutral pH. The demineralized *G. orientalis* was directly used for the next step without further purification or drying.

Step 2, deproteination: after step 1, the resulting exoskeleton was placed in an aqueous 2M NaOH solution. The solution was then warmed to 95 °C and held at this temperature for 36 h. During this period, the solution rapidly transitioned to a black colour. The NaOH solution was refreshed every hour until the solution maintained a clear brown appearance. The liquid phase was then removed and the *G. orientalis* specimen was rinsed with fresh water until reaching a neutral pH. The *G. orientalis* specimen was directly used for the next step without further purification or drying.

Step 3, bleaching: lastly, the exoskeleton was bleached using an aqueous H_2_O_2_ solution (50% w/w) at room temperature for 4 h. The *G. orientalis* was then washed with water and acetone. The treated *G. orientalis* was dried in oven (60 °C), yielding 1.7 g of Chitin (yield 16%).

### 2.2. Chitin Deacetylation (Chitosan Preparation)

One gram of dry chitin was rehydrated in a 50% (w/w) NaOH water solution. The solution was then warmed (95 °C) overnight. The liquid phase was removed and the solid was washed with fresh water until reaching a neutral pH. The deacetylated chitin (chitosan) was then washed with acetone and dried in an oven (60 °C), yielding 0.8 g (yield: 80 %) of chitosan.

### 2.3. Surface Characterization

All surface characterizations were performed on both *G. orientalis* and commercially available shrimp chitosan. All observations were performed 3 times to obtain standard deviation.

#### 2.3.1. Electronic Microscopy

SEM observations were carried out using Phenom ProX scanning electron microscope. Samples were observed with gold coating at an accelerating voltage of 5 and 10 kV. The samples were coated using Q150R S Sp.

#### 2.3.2. Infrared Measurement

Infrared measurements were carried out using a Spectrum Two FT-IR spectrometer from Perkin Elmer with a universal ATR accessory. The measurements were performed between 4000 cm^−1^ and 500 cm^−1^.

#### 2.3.3. Determination of the Deacetylation Degree of the Chitosan

Dried chitosan (0.1 g) was dissolved in 30 mL of 0.1 M HCl acid. When chitosan was completely dissolved, the solution was titrated with a 0.1 M NaOH solution.

Deacetylation degree of chitosan [48,49] was calculated using the following formula:
$$\mathrm{Deacetylation}\;\mathrm{degree}\;(\%)\;=2.03\times \frac{V2-V1}{m+0.0042*\left(V2-V1\right)}$$ where m is sample mass (g), V1 and V2 are volumes of NaOH solution corresponding to the deflection points for HCl and chitosan hydrochloride, respectively, 2.03 is a coefficient resulting from the molecular weight of the chitin monomer unit and 0.0042 is a coefficient resulting from the difference between the molecular weights of the chitin and chitosan monomer units.

#### 2.3.4. TGA Measurement

Weight measurements were performed using a 403 Aëolos Quadro quadrupole mass spectrometer from Nezsch. The samples were warmed from 40 °C to 650 °C at a heating rate of 20 °C min^−1^, and then the temperature was held at 650 °C for a period of 4 h.

## 3. Results

### 3.1. Chitin and Chitosan Preparation

Chitin extraction and chitosan preparation can be carried out following various methods based in either chemical or enzymatic strategies [50,51,52,53]. If the enzymatic strategy can be reported as a green way to produce chitosan, the chemical strategy is well established and allows production of chitosan with a high deacetylation degree. In this work, we employ a simple chemical strategy used for chitin and chitosan extraction [54]. In our strategy, we extract chitin in three straight-forward steps: (1) demineralization, (2) deproteination and (3) bleaching. Following the three-step chitin extraction, chitosan can then be obtained using a one-step strategy. As stated, chitin extraction begins with demineralization, which is achieved by immerging dry insect specimens in a 1M HCl solution. The mixture is then warmed at 95 °C for 2 h, after which the liquid phase is removed, and the demineralized insect cuticle is thoroughly washed with water until achieving a neutral pH. The resultant material was directly used for the next step, deproteination, without drying. For deproteination, the material was immersed in 2M NaOH solution at 95 °C for 36 h. The NaOH solution turned black rapidly and was therefore refreshed regularly (every hour) until the solution stopped changing colour. After a 36-hour period, the liquid phase was removed and the solid was thoroughly washed with water until reaching a neutral pH. The bleaching step was performed to remove any coloration of the chitin, using hydrogen peroxide (50% w/w) and washing with acetone to remove any potential residual trace fat. The final isolated solid is described as chitin (Figure 2A).

The final yield of isolated, dried chitin was 16%, a value consistent with other reports in the literature [41]. Chitosan preparation from the extracted chitin was then performed in one step by immersing dried chitin in a 50/50 w/w aqueous NaOH solution and then heating the mixture to 95 °C overnight. After this period, the liquid was removed and the solid was thoroughly washed with water until reaching a neutral pH. Finally, the solid was washed with acetone and dried (yield 80%). The obtained compound was a poly (β-(1-4)-D-glucosamine) with random presence of N-acetyl-D-glucosamine, e.g. chitosan (Figure 2B). This compound was easily dissolved in a diluted HCl solution by forming the corresponding hydrochloride salt (Figure 2C).

### 3.2. Macroscopic Observations

During the chitin extraction, important macroscopic modifications were observed. Obviously, the demineralisation and the deproteination had a dramatic impact on the specimens. For example, most major parts of the specimens (elytra, pronotum etc.) detach during this process. However, other more subtle changes were observed, particularly with regards to colour.

*G. orientalis* is known as a black and white giant beetle, as shown in Figure 3A [55,56]. The appearance of the specimen did not change significantly after demineralisation; however, after deproteination, the *G. orientalis* appearance was modified dramatically to reveal an iridescent aspect. At this point, the majority of the body was maroon in color, with gold or intense green reflections depending on the observation angle (Figure 3B). This is not surprising, as iridescent colours are a common property of beetles [57,58], and this iridescent effect is attributed to the chitin network. Therefore, it is not surprising that after deproteination and exposure of the chitin network, the iridescent properties are strongly evident. While it is well established that this metallic coloration can be removed from beetle by applying physical or chemical treatments, there are limited demonstrations that this class of optical properties can be exposed from species with non-metallic aspect [59,60,61]. After bleaching, the iridescent effect was preserved, demonstrating that bleaching does not disturb the chitin network; however, there was an associated colour change to a shiny pearl hue (Figure 3C). Unsurprisingly, after drying, the network changed dramatically, and no more iridescent colour could be observed (Figure 3D); this change in colour can probably be linked to the modification of the chitin network after drying.

### 3.3. Microscopic Observations of G. orientalis Surface

If macroscopic variations in the specimen evolve during chitin extraction, it is reasonable to consider the possibility that there are also microscopic changes. For this reason, each stage of chitin extraction and chitosan preparation was observed using scanning electronic microscopy (SEM). The native surface morphology of *G. orientalis* has been previously described in the literature [47] and it was revealed that the white domains of the shell have large fibre-like elements (Figure 4A) and the black sections are relatively smooth (Figure 4B).

However, after demineralisation, we observed here that the white regions had an increasingly disordered morphology (Figure 5A), while the black domains remain, still, relatively smooth (Figure 5B). These results are consistent with the previously described macroscopic observations (Figure 3), as the general macroscopic morphologies and surface patterns are not significantly altered after demineralisation.

After deproteination, the black and white sections of the shell no longer have distinguishable microscopic morphologies. While there are a few regions of the surface that can be characterized as having irregular morphology (Figure 6A), the majority of the surface is characterized by a well-defined chitin network, as shown Figure 6B.

Microscopic observations made after bleaching yield similar results to Figure 6, further confirming the macroscopic observations. After bleaching, very few sections of the surface have minimal roughness (Figure 7A), while the majority of the surface again yields the chitin network. This network likely contributes to the iridescent effect observed on the macroscopic scale (Figure 7). Furthermore, with cross-section microscopy, it is possible to observe the layer-by-layer structure of the surface (Figure 7C,D).

After deacetylation, the entire surface is covered by a well-defined network (Figure 8A,B); after deproteination and bleaching, the layer-by-layer structure of the chitin is preserved and can be observed using cross-section microscopy (Figure 8C,D).

### 3.4. G. orientalis Chitosan Characterisations

To confirm the formation of chitosan, material characterisations such as infrared spectroscopy were employed (Figure 9).

In the FT-IR spectra, all characteristic bands from chitosan are observed. This includes strong bands at 3340–3380 cm^−1^ and 3280–3293 cm^−1^, attributed to stretching of O-H and N-H groups, respectively. Furthermore, the band at 2846–2886 cm^−1^ is attributed to CH_2_ vibration. Not surprisingly, the C=O band of the amide group is weak but can be observed at 1642–1660 cm^−1^, consistent with the partial deacetylation of chitin. Lastly, the bending band vibration band from N-H is observed at 1577–1588 cm^−1^. Compared to FT-IR spectra obtained from shrimp chitosan, no significant difference is noted.

The deacetylation degree was estimated using a titration method [48,49]. In this method, chitosan is dissolved in a HCl (1M) solution. In this condition, all free amino groups form an ammonium chloride and any excess HCl remains in solution. The HCl/chitosan ammonium chloride solution is then titrated using NaOH (1M). The titration curve shows two deflections. The first one corresponds to HCl and the second one to chitosan hydrochloride (Figure 10).

Using this approach, the deacetylation degree was determined to be 78.3 ± 1.5% for *G. orientalis*. A similar calculation was performed for the shrimp chitosan, indicating 74.8 ± 0.4% deacetylation. Both of these results are consistent with data published in the literature for shrimp or for various beetles [39].

Ash content is another important parameter used to evaluate chitosan quality. Ash content dramatically changes the viscosity and viscoelastic behaviour of materials formed from chitosan. For example, to form hydrogels, the ash content should be as low as possible (below 5%). The ash content of *G. orientalis* was investigated using thermogravimetric analysis (Figure 11).

Thermogravimetric analysis reveals that chitosan from *G. orientalis* has an ash content of 9.1%, which, compared to the ash content for shrimp chitosan (11.3%), is relatively similar. The isolated chitosan in this study already presents a low ash content but should be purified for further applications.

## 4. Conclusions

In conclusion, we report the use of *G. orientalis* for the preparation of chitin and chitosan using a chemical treatment. After a multi-step process including demineralisation and deproteination, the chitin is obtained from *G. orientalis* with a yield of 16%. This extraction procedure for chitin reveals the hidden macroscopic optical effect of *G. orientalis*. Furthermore, SEM observations reveal the microscopic evolution of the surface morphology with the extraction of the chitin network, which reveals an organized surface with limited rough, disordered domains. Chitosan is obtained after a deacetylation step of the chitin, with a yield of 80%, a deacetylation degree of 78.3% and 9.1% ash content. Deacetylation degree and ash content are similar for our *G. orientalis* chitosan and commercially available shrimp chitosan. Furthermore, the *G. orientalis* chitosan has very similar properties compared with commercially available shrimp chitosan, as revealed by infrared spectroscopy. As a first observation, this work shows the possibility to extract chitin and chitosan from large-bodied beetles. In the future, various other large-bodied specimens from other species and genera should be investigated. Additionally, the prepared chitosan produced here will be investigated further for hydrogel elaboration, and the physicochemical properties of these gels will be investigated. The main final objective of this work will be to consider beetle chitosan for applications in 3D printing or biomedical applications.

## Figures and Tables

**Figure 1 biomimetics-05-00015-f001:**
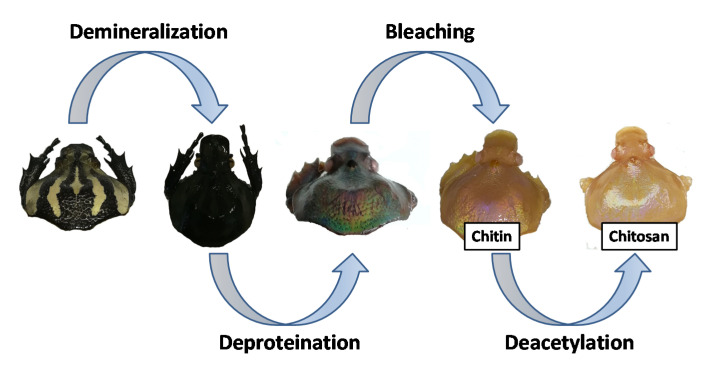
General concept for chitosan preparation from *G. orientalis*.

**Figure 2 biomimetics-05-00015-f002:**
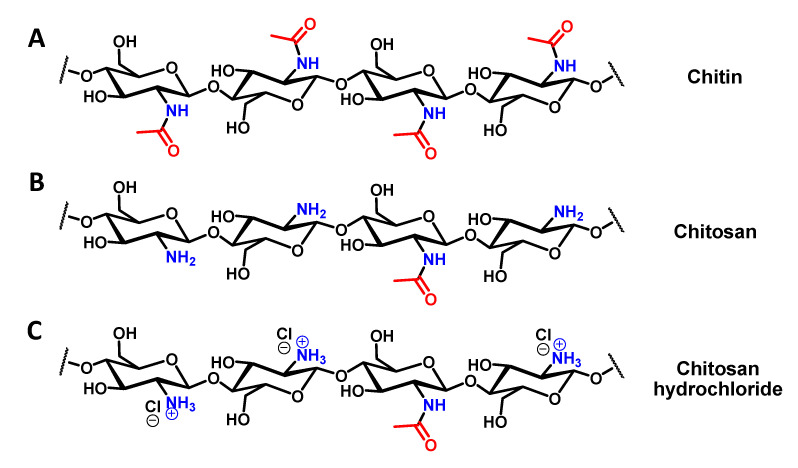
Chemical structures of the formed molecules. (**A**) Chemical structure of chitin; (**B**) Chemical structure of chitosan; (**C**) Chemical structure of chitosan hydrochloride.

**Figure 3 biomimetics-05-00015-f003:**
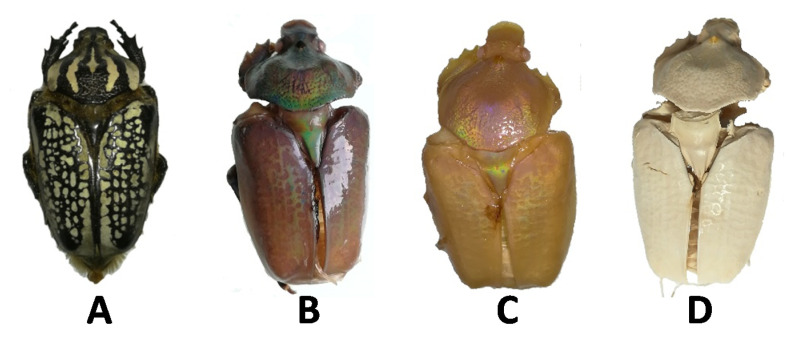
Example of *G. orientalis* morphological evolutions. (**A**) Starting specimen; (**B**) Specimen after deproteination; (**C**) Specimen after bleaching; (**D**) Specimen after drying.

**Figure 4 biomimetics-05-00015-f004:**
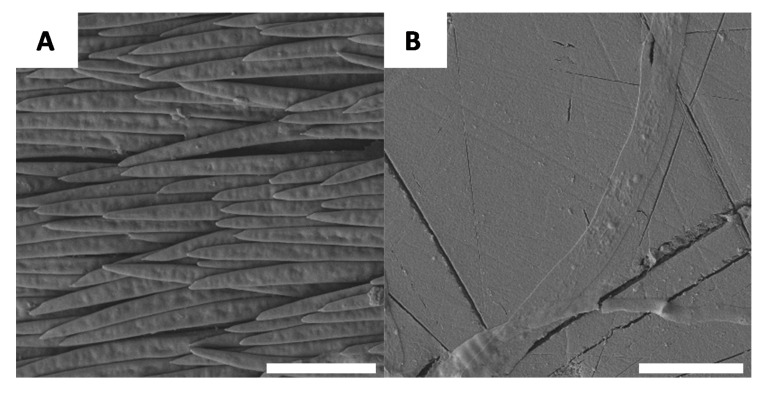
Example of native surfaces for *G. orientalis*, white (**A**) and black (**B**) part (Scale bar = 10 µm).

**Figure 5 biomimetics-05-00015-f005:**
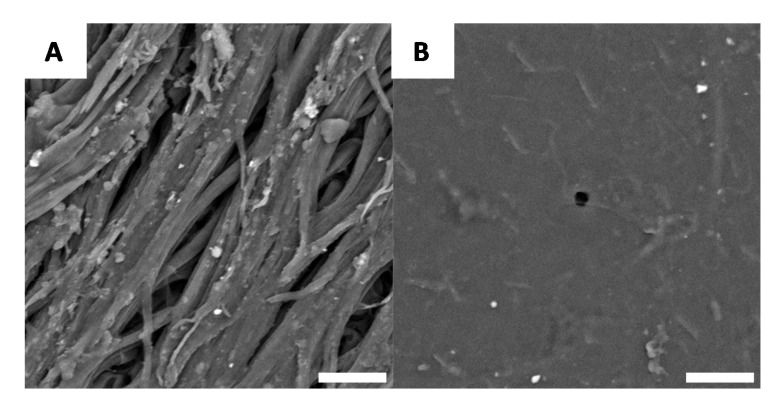
Example of surface morphology after demineralisation, for white (**A**) and black (**B**) part (Scale bar = 10 µm).

**Figure 6 biomimetics-05-00015-f006:**
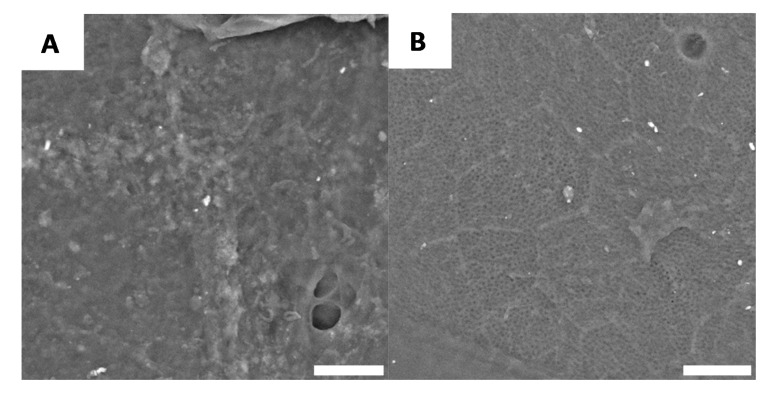
Examples of morphology observed for *G. orientalis* after deproteination (Scale bar = 10 µm). (**A**) Example of observed irregular morphology; (**B**) Example of chitin network.

**Figure 7 biomimetics-05-00015-f007:**
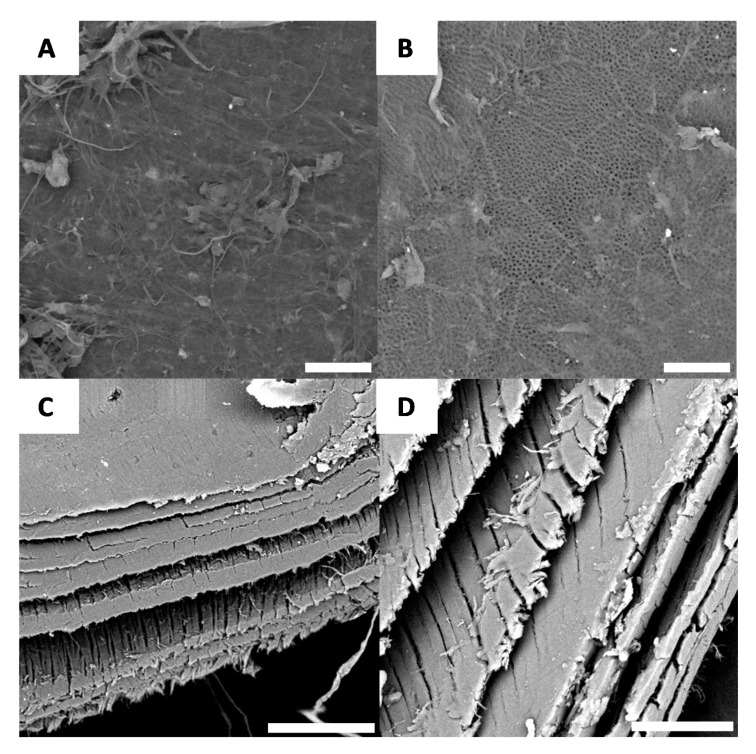
Examples of morphology observed for *G. orientalis* after bleaching (**A** and **B**) (Scale bar = 10 µm); Cross-section observations for *G. orientalis* after bleaching (**C** and **D**) (Scale bar = 80 µm).

**Figure 8 biomimetics-05-00015-f008:**
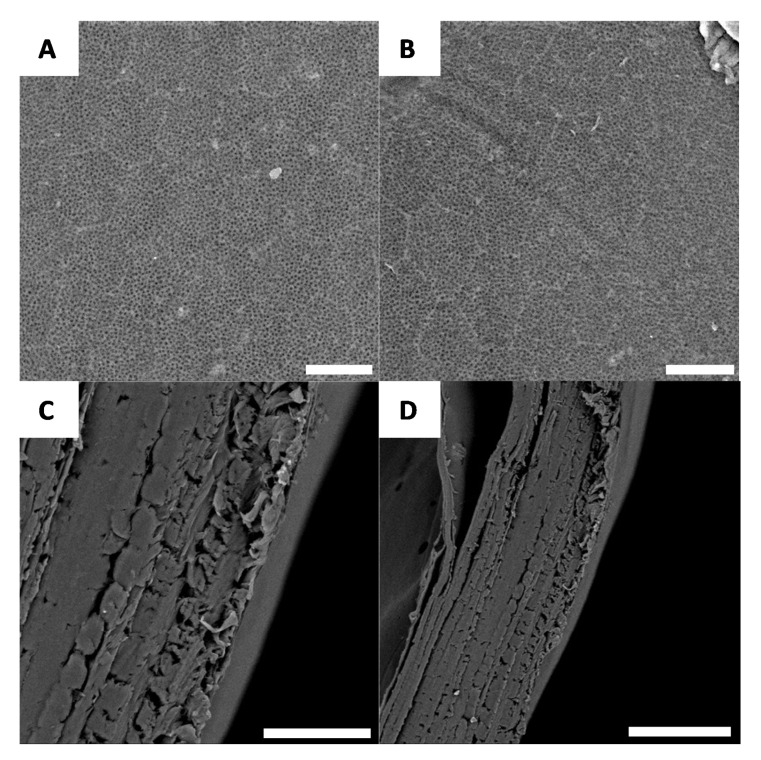
Examples of morphology observed for *G. orientalis* chitosan (**A** and **B**) (Scale bar = 10 µm); Cross-section observations for *G. orientalis* chitosan (**C** and **D**) (Scale bar = 80 µm).

**Figure 9 biomimetics-05-00015-f009:**
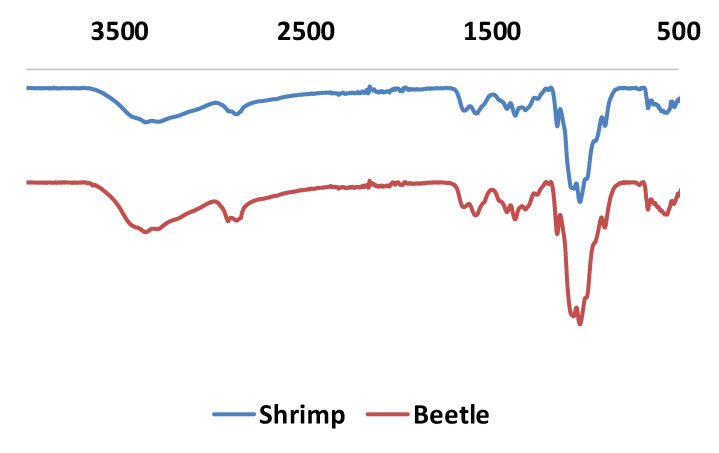
Infrared measurements for chitosan from shrimp (blue) and *G. orientalis* (red).

**Figure 10 biomimetics-05-00015-f010:**
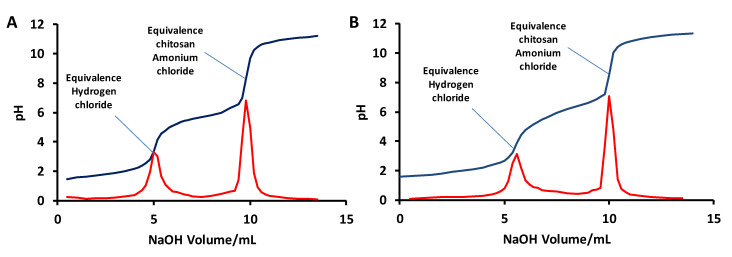
Example of titration curve for *G. orientalis* (**A**) and shrimp (**B**) chitosan deacetylation degree determination.

**Figure 11 biomimetics-05-00015-f011:**
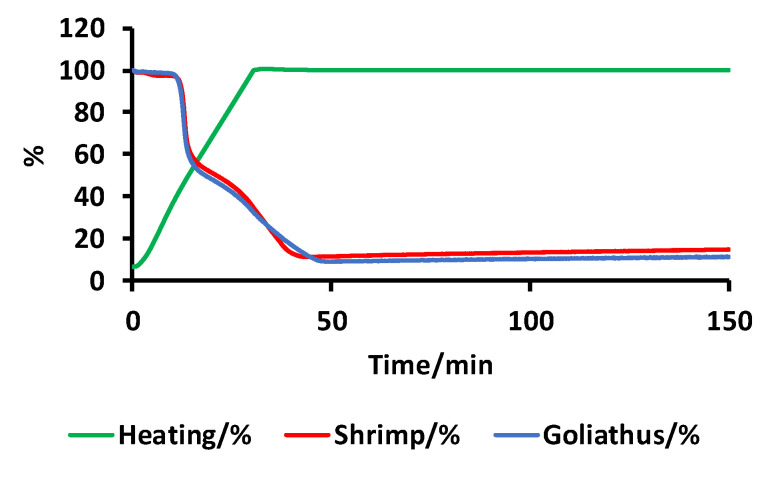
Thermogravimetric measurements for chitosan from *G. orientalis* (blue) and shrimp (red).

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
