# Peer review of "Chitosan Extraction from Goliathus orientalis Moser, 1909: Characterization and Comparison with Commercially Available Chitosan"

_biomimetics, 2020, doi:10.3390/biomimetics5020015_

Round 1

Reviewer 1 Report

The author described a method to extract chitosan from Goliath beetle. I suggest a few points to be addressed before it can be published. 

  1. In the introduction, the authors may explain a little bit more about the rationale of extracting chitosan from Goliath beetle rather than commonly used seafood waste. 
  2. In section 2.2, the author calculated that the yield is 16 %. Are there impurity in the extracted materials? I would suggest the author to characterize the purity of the obtained materials/chemicals. 
  3. The microstructures of the black and white region is interesting. Did the author observe any difference in deacetylation degree for those two regions
  4. What are the advantages of chitosan derived using the author's method compared with other ways? The author can demonstrate the advantages in specific applications. 

Author Response

  1. In the introduction, the authors may explain a little bit more about the rationale of extracting chitosan from Goliath beetle rather than commonly used seafood waste.

Authors answer: The author added an additional sentence to the introduction

  1. In section 2.2, the author calculated that the yield is 16 %. Are there impurity in the extracted materials? I would suggest the author to characterize the purity of the obtained materials/chemicals. 

Authors answer: In the present work, the group calculated the crude yield. Few impurities can be present. Elemental analysis can be performed to give more details on purity. Unfortunately, due to the special situation (COVID-19) our laboratory and analysis department are closed. I won’t be able to perform additional measurements within the period given for manuscript corrections (10 days). If possible, the group can add this measurement before the final, published version.

  1. The microstructures of the black and white region is interesting. Did the author observe any difference in deacetylation degree for those two regions?

Authors answer: The authors thank the reviewer for this interesting question. No difference can be observed between black and white part after the chitin extraction and as consequence after deacetylation.

  1. What are the advantages of chitosan derived using the author's method compared with other ways? The author can demonstrate the advantages in specific applications. 

Authors answer: In this work, we report the use of regular chemical strategy for chitin extraction. The authors have added a sentence to emphasize this point.

Reviewer 2 Report

1- Title of the paper should be clear and more focused.

2- The novelty and objective of the manuscript need to be highlighted clearly. 

3- In this paper author only used two charecterization technique to evaluate newly obtained chitosan. It will be nice if author could include solid state NMR and biocompatibility of the new chitosan.

4- In Figure 9 caption and spectra are not equal, please correct it

5- In Figure 10 author only gave titration curve for G. orientalis chitosan deacetylation degree determination, it would have been good if, author could compare with commercially available chitosan shrimp.

6- Try to keep both the specta in one figure 10 instead of Figure 10A and 10B, it will be easier for the reader to compare, if two spectra will be in one figure.

7- The overall results and discussion needs to be improved.

8- Authors can improve conclusion which can highlight major finding and the prospective of the study.

9-Authors can insert some related recent references.

Author Response

1- Title of the paper should be clear and more focused.

Authors answer: The title has been modified and clarified

2- The novelty and objective of the manuscript need to be highlighted clearly. 

Authors answer: The authors have added sentences to clarify this point and emphasize the novelty of the presented work in the introduction.

3- In this paper author only used two charecterization technique to evaluate newly obtained chitosan. It will be nice if author could include solid state NMR and biocompatibility of the new chitosan.

Authors answer: The group agree with the reviewer, solid state NMR or liquid state NMR (using D2O and Acetic acid D4) can be performed. Unfortunately, due to the special situation (COVID-19) ower laboratory and NMR department are closed. I’ll not be able to perform addition measurements in the corrections delay (10 days). If possible, the group will add this measurement before the final version.

4- In Figure 9 caption and spectra are not equal, please correct it

Authors answer: Done

5- In Figure 10 author only gave titration curve for G. orientalis chitosan deacetylation degree determination, it would have been good if, author could compare with commercially available chitosan shrimp.

Authors answer: Done

6- Try to keep both the specta in one figure 10 instead of Figure 10A and 10B, it will be easier for the reader to compare, if two spectra will be in one figure.

Authors answer: In the initial submission, Figure 10 contains only 1 curve, I assume that the reviewer is referring to figure 11. The suggested modification has been made.

7- The overall results and discussion needs to be improved.

Authors answer: Discussion have been significantly edited and improved.

8- Authors can improve conclusion which can highlight major finding and the prospective of the study.

Authors answer: Conclusion has been significantly edited and improved.

9-Authors can insert some related recent references.

Authors answer: Recent references, including some from 2020 have been added.

Round 2

Reviewer 2 Report

  1. The title of the manuscript can be Chitosan extraction from Goliathus orientalis Moser: Characterization and comparison with commercially
    available shrimp chitosan.
  2. Only IR, Titration and TGA will not be enough to justify the objective.

Author Response

1- The title of the manuscript can be Chitosan extraction from Goliathus orientalis Moser: Characterization and comparison with commercially available shrimp chitosan.

Author answer: The title has been modified as required.

2- Only IR, Titration and TGA will not be enough to justify the objective.

Author answer: In response to Reviewer's 2 critiques, we acknowledge that additional experiments could supplement and strengthen the conclusions of this manuscript. Unfortunately, due to the Covid-19 pandemic, we are currently restricted from accessing our laboratory facilities until at least May 11th (and potentially later, depending on the development of this global issue). Therefore, we cannot provide additional experiments in the time allotted from the editorial staff. An extension from the editorial staff would be the only way to address this issue.

This measurement can be easily performed in normal conditions. The problem comes from COVID-19 pandemics.

In my opinion it is not very important or necessary for the main idea of the article. In the literature chitosan NMR is often not presented

For Chitosan as for other polymers, NMR will present broad peaks and will not be as informative as for small molecules. By comparing on the acetyl CH3 and a second signal it is possible to get the deacetylation degree using 1H NMR. But we already determinate the deacetylation degree using the titration. 13C NMR will provide broad signals around 105.01ppm, 84.14 ppm, 76.64 ppm, 74.41 ppm, 61.95 ppm, 56.14 ppm and 23.79 ppm as reported for commercial Chitosan. This observation is consistent with the sugar backbone of chitosan. But it will not provide significant improvement for the paper in my opinion.

XRD, Elemental analysis and biological assay will be performed in the future to use our chitosan in bio sourced materials (for 3D printing or biomedical application). But it is a complete other work that the group will be glad to submit in biomimetics as soon as we performed it.

If the reviewer insists and the NMR is realy needed, it can be performed but it will takes time before the lab re-open.